# Deep-Learning-Based Automatic Segmentation of Parotid Gland on Computed Tomography Images

**DOI:** 10.3390/diagnostics13040581

**Published:** 2023-02-04

**Authors:** Merve Önder, Cengiz Evli, Ezgi Türk, Orhan Kazan, İbrahim Şevki Bayrakdar, Özer Çelik, Andre Luiz Ferreira Costa, João Pedro Perez Gomes, Celso Massahiro Ogawa, Rohan Jagtap, Kaan Orhan

**Affiliations:** 1Department of Dentomaxillofacial Radiology, Faculty of Dentistry, Ankara University, Ankara 06000, Turkey; 2Dentomaxillofacial Radiology, Oral and Dental Health Center, Hatay 31040, Turkey; 3Health Services Vocational School, Gazi University, Ankara 06560, Turkey; 4Department of Oral and Maxillofacial Radiology, Faculty of Dentistry, Eskisehir Osmangazi University, Eskişehir 26040, Turkey; 5Eskisehir Osmangazi University Center of Research and Application for Computer-Aided Diagnosis and Treatment in Health, Eskişehir 26040, Turkey; 6Division of Oral and Maxillofacial Radiology, Department of Care Planning and Restorative Sciences, University of Mississippi Medical Center School of Dentistry, Jackson, MS 39216, USA; 7Department of Mathematics-Computer, Faculty of Science, Eskisehir Osmangazi University, Eskişehir 26040, Turkey; 8Postgraduate Program in Dentistry, Cruzeiro do Sul University (UNICSUL), São Paulo 01506-000, SP, Brazil; 9Department of Stomatology, Division of General Pathology, School of Dentistry, University of São Paulo (USP), São Paulo 13560-970, SP, Brazil; 10Department of Dental and Maxillofacial Radiodiagnostics, Medical University of Lublin, 20-093 Lublin, Poland; 11Ankara University Medical Design Application and Research Center (MEDITAM), Ankara 06000, Turkey

**Keywords:** artificial intelligence, deep convolutional neural network, salivary glands, U-net, computed tomography

## Abstract

This study aims to develop an algorithm for the automatic segmentation of the parotid gland on CT images of the head and neck using U-Net architecture and to evaluate the model’s performance. In this retrospective study, a total of 30 anonymized CT volumes of the head and neck were sliced into 931 axial images of the parotid glands. Ground truth labeling was performed with the CranioCatch Annotation Tool (CranioCatch, Eskisehir, Turkey) by two oral and maxillofacial radiologists. The images were resized to 512 × 512 and split into training (80%), validation (10%), and testing (10%) subgroups. A deep convolutional neural network model was developed using U-net architecture. The automatic segmentation performance was evaluated in terms of the F1-score, precision, sensitivity, and the Area Under Curve (AUC) statistics. The threshold for a successful segmentation was determined by the intersection of over 50% of the pixels with the ground truth. The F1-score, precision, and sensitivity of the AI model in segmenting the parotid glands in the axial CT slices were found to be 1. The AUC value was 0.96. This study has shown that it is possible to use AI models based on deep learning to automatically segment the parotid gland on axial CT images.

## 1. Introduction

Salivary glands are important exocrine organs of the human body, responsible for the production of saliva as well as various digestive enzymes. Human salivary glands are divided into major and minor glands according to their size and function. The major salivary glands are defined as the parotid, submandibular, and sublingual glands [1,2]. The parotid gland is the largest in size, responsible for producing 60 to 65% of the oral cavity’s total saliva, and envelops the mandible’s ramus [3,4]. The facial nerve subdivides the parotid gland into superficial and deep lobes [5]. In 2017, the World Health Organization proposed a classification with more than 30 types of salivary gland tumors categorized as being either malignant or benign histological subtypes. Benign salivary gland tumors constitute approximately 6% of tumors diagnosed in the head and neck region [1,2]. Salivary gland tumors can originate from distinct types of glandular cells, and they exhibit considerable variances in their clinical, pathological, and biological characteristics. The current treatment options are multimodality therapy, chemotherapy, radiation therapy, and surgical resection [6]. The malignant or benign characteristics of the salivary gland tumor are important in terms of the prognosis and treatment options, since malignant tumors require a more invasive operation [7,8]. Improvements in salivary gland imaging, consistent with the histopathological findings, will contribute to the relevant clinical decision [9,10].

Several imaging techniques can be adopted to reveal the status of the parotid glands, each with its own advantages and limitations. The magnetic resonance imaging (MRI) and computed tomography (CT) techniques are the primary methods for evaluating the parotid gland anatomically, pathologically, and structurally, by enabling the cross-sectional evaluation of the salivary glands [11,12,13]. CT is proposed for cases where an inflammatory condition such as sialectasis, abscess, stone, and acute inflammation is suspected, and when MRI is contraindicated. Nevertheless, MRI is the preferred imaging technique in patients with a high suspicion of malignancy. In addition, ultrasonic imaging can be beneficial in pediatric and pregnant patients for an initial investigation, particularly in cases involving lesions of the parotid gland’s superficial lobe [14].

Progress in digital imaging has paved the way for implementing various artificial intelligence (AI) tools for segmenting, detecting, and classifying the anatomical and pathological structures [15,16]. Currently, the practice of radiology benefits significantly from AI applications. Implementing such tools can be highly beneficial in removing the burden of performing certain tasks repeatedly including segmenting organs or nerves or for extracting the quantitative data that are more beneficial, thus enabling clinicians to increase their focus on attempting to solve complicated clinical issues [17,18]. Still, there are many problems, such as the need for large datasets and training, regulation issues, and medicolegal responsibility, which are suggested as barriers to the efficient application of AI in radiologists’ normal practice. For the success of a developed AI model, the use of quality data in education and the correct labeling process are both important [19,20].

Image segmentation is the subject of various fields such as transportation, architecture, and medical imaging. Traditional segmentation methods, such as boundary extraction, threshold-based segmentation, and region-based segmentation can be adopted in manual segmentation of the medical images [21,22]. However, manual segmentation requires expertise and is a time-consuming process. In the deep learning approach, features are extracted by algorithms by establishing multilayered mathematical models. Thus, developers can benefit from the advantage of using big data in model training [22,23]. Convolutional neural network (CNN) algorithms have received attention for their success in image processing tasks. U-Net is an architecture developed for image segmentation. The basic structure consists of contraction and expansion paths, which are almost symmetrical, resulting in a u-like shape [21,22,23]. Deep learning can be utilized in U-Net algorithms, and its better performance than its competitors using a limited dataset makes this architecture popular in segmentation tasks in the medical field where data are limited [21].

Recently, the deep learning method has been utilized extensively, especially in medical image processing where segmentation is needed [16,24]. Segmentation from head and neck CT images has been performed with the deep learning method [25,26]. In 2014, Yang et al. proposed a system based on atlas registration and a support vector machine model for automated segmentation of the parotid gland using MR images. Fifteen patients with head and neck radiotherapy (42 MRI data) were included, and the difference between the model and the human tracings was reported as 7.98% and 8.12% for the left and the right parotid, respectively [27]. In 2018, Močnik et al. developed an automatic multimodal method for segmentation of the parotid glands from a CT and MRI pair of patient data. Elastix and ANTs tools were employed to register the MRI image to the CT, and the CNN model was implemented using Microsoft Cognitive Toolkit. The researchers compared the results of the proposed multimodal model with the CT-only modality and reported a Dice overlapping coefficient of 78.8% for the first and 76.5% for the latter approach [28]. Hänsch et al. developed a U-Net based system for segmenting the parotid from CT images that were two-dimensional, three-dimensional, and in a two-dimensional ensemble mode, in 2019. In total, 254 head and neck CT scans from two different clinical sites were selected, and in addition to the models’ performance for segmentation, the number of the training samples needed was also investigated. The authors reported a mean Dice similarity of 0.83 for all three models, and increasing the training cases to more than 250 did not increase the Dice coefficient significantly [29].

This study aims to develop a deep convolutional neural network (dCNN) algorithm based on U-Net architecture and to evaluate the model’s performance in the automatic segmentation of the parotid glands on axial-CT images.

## 2. Materials and Methods

### 2.1. Study Design

A U-net based algorithm was developed using the Pytorch library for the automatic segmentation of the parotid gland in axial slices of head and neck CT images (CranioCatch, Eskisehir-Turkey). All procedures performed in studies involving human participants were in accordance with the ethical standards of the institutional and/or national research committee and with the 1964 Helsinki declaration and its later amendments or comparable ethical standards. The study protocol was approved by the Non-interventional Clinical Research Ethics Board of The University of Campinas (UNICAMP) with the decision number 79765917.5.0000.5404 (decision date 18 March 2018, meeting number 2.553.836).

### 2.2. Study Data

In this retrospective study, 30 anonymized CT datasets were selected from the archive of the Radiology Department of the Faculty of Medical Sciences, University of Campinas (UNICAMP). Samples with clearly visible parotid glands bilaterally were included, while images with a gross anomaly and artifacts on the parotid gland were excluded. Radiographic data were acquired by a 16-slice CT scanner (Siemens Somatom Sensation 16, Forcheim, Germany) with the constant parameters of 0.6 mm detector collimation, 120 kVp tube voltage, 0.6 s gantry rotation time, 1.5 mm reconstructed section thickness, and 1 mm reconstruction intervals. The patient data in three-axes (sagittal, coronal, and axial) were reconstructed into volumetric data and exported in Digital Imaging and Communication in Medicine (DICOM) file format. The resulting DICOM files were imported to Pydicom (https://pydicom.github.io/datasets (accessed on 1 June 2022)) software, and in total, 931 axial-CT images with a unilateral or bilateral appearance of the parotid gland were exported in Joint Photographic Experts Group (JPEG) format.

### 2.3. Ground Truth Labeling

The CranioCatch Annotation Tool (CranioCatch, Eskisehir, Turkey) was developed with polygonal box segmentation technique for labeling of the parotid glands on the axial CT images. The ground truth was determined by the consensus of two experts in oral and maxillofacial radiology (I.S.B. with 11 years’ experience and M.O. with 2 years’ experience).

### 2.4. Data Split

The 931 axial images were resized to 512 × 512 pixels. The dataset was separated into the training (80%), validation (10%), and testing (10%) groups randomly.

Training group: 745 (1445 labels);Validation group: 93 (178 labels);Testing group: 93 (184 labels).

### 2.5. Development of the U-Net Based dCNN Model

The U-net based automated parotid segmentation algorithm was developed in the Python environment (v.3.6.1; Python Software Foundation, Wilmington, DE, USA) using the PyTorch library. The model was trained for 700 epochs with learning rate of 0.00001. Mathematical processing in the model’s training was performed with a Dell PowerEdge T640 Calculation Server (Dell Inc., Round Rock, TX, USA), Dell PowerEdge T640 GPU Calculation Server (Dell Inc., Round Rock, TX, USA), and a Dell PowerEdge R540 Storage Server (Dell Inc., Round Rock, TX, USA) in the Eskisehir University Dentistry Faculty Dental-AI Laboratory, (Appendix A), (Figure 1).

### 2.6. Statistics for the Model’s Performance

The model’s performance in the automated segmentation of the parotid glands on the axial CT images was evaluated with the F1-score, the precision, the sensitivity, and the area under curve (AUC) values. The model’s result was considered successful if the prediction and the ground truth intersected by more than 50% in each individual image slice. The true positive (TP), false positive (FP), and false negative (FP) results were determined for calculating the performance metrics. The definitions and the formulas for calculating the model’s performance are described below:True positive (TP): At least 50% of the pixels intersect between the automatic segmentation algorithm and the ground truth;False positive (FP): At least 50% of the pixels of the automatic segmentation algorithm do not intersect with the ground truth;False negative (FN): At least 50% of the pixels of the ground truth do not intersect with the results of the automatic segmentation algorithm;Sensitivity (Recall, True positive rate (TPR)) = TP⁄((TP + FN));Precision (Positive predictive value (PPV)) = TP⁄((TP + FP));F1-Score = 2TP⁄((2TP + FP + FN)).

## 3. Results

The U-Net based algorithm (CranioCatch, Eskisehir-Turkey) predicted the pixels of the parotid glands with more than 50% intersection in all samples (Figure 2). The values of the F-measure, precision, and sensitivity were all determined to be 1.0 in terms of segmenting the parotid gland axial slices of CT images successfully (Table 1). The Area Under Curve (AUC) value was found to be 0.96 (Figure 3 and Figure 4).

## 4. Discussion

As new developments occur in terms of deep learning and neural techniques, artificial intelligence is being increasingly integrated into the field of medicine, and artificial intelligence has been used to solve clinical problems. Recently, at the same time as deep learning techniques are being used in the medical field, its application in dentistry has also increased. In the current study, the technique employed offers a comprehensive training approach to optimize the usage of datasets that have been partly annotated for the purpose of segmenting organs. Segmenting organs with precision and reliability can help to improve clinical applications including computer-aided detection, treatment, and surgical procedures. Organ segmentation also has the potential to be a critical factor in educating dental students [16,30,31]. Our study enables the segmentation of distinct IT images using a single network. To take advantage of the data from large scale datasets, previous researchers have adopted semi-supervised approaches in which the data were labeled weakly or potentially had no labels. This study is supplementary to previous approaches, and it is possible to amalgamate it with semi-supervised learning to assist with overcoming the issue of data need when segmenting organs. The findings of this study show that there were minimal differences in terms of the segmentation performance when training was performed on a large-scale dataset containing clinical quality references compared to a dataset that was smaller in size with curated quality references. In the future, an important additional step will involve the clinical qualitative assessment of the clinical admission of the contours that deep learning generates. It has been found the networks that are deeper with an increased number of parameters are also capable of consubstantiating a greater amount of data and facilitating additional improvements in the segmentation performance using additional samples. Furthermore, the performance of the deep learning techniques was more robust and had less variance compared to methods based on model- or atlas-based approaches with regard to the segmentation task. This could be due to the fact that the learned attributes could be representative of a broad anatomical diversity with no previous assumptions, and training may also have been conducted on a dataset with a larger size compared with the techniques used in the task [29,32].

In a study in which segmentations of five different internal organs were evaluated using the U-net algorithm in 2020, the accuracy values for these organs were determined to be 0.959, 0.813, 0.595, 0.900, and 0.911, respectively [33]. In our study, the accuracy value was found to be 1.0 for the parotid gland. A study conducted in 2016 focused on designing and training a 3D convolutional neural network for automatic detection of the liver, where the training dataset comprised 151 CT images, the validation dataset included 20 images, and the testing group included 10 images. In the results of this study, the average accuracy value for the liver segmentation was found to be 97.6% [34]. Again, similar to our study, in a study using the U-net algorithm, CT images of COVID-19 patients were evaluated and the values for the sensitivity, precision, and F1-score were calculated as 0.8, 0.82 and 0.81, respectively. In the same study, it was shown that the results could be further improved by adding various modules to the U-net algorithm [35]. In another study using a fully connected network, which is a somewhat similar method, photographs of skin lesions were evaluated, and the F1-score and sensitivity values were found to be 0.912 and 0.918, respectively [36]. In another study performed with CT images of individuals diagnosed with COVID-19, the sensitivity and F1-score values were found to be 0.439 and 0.534, respectively, unlike our study and other similar studies [37]. In a study conducted in 2021 comparing human and CNN-based diagnosis, 855 CT images were used for training and validation and 256 true-positive, 279 false-positive, and 114 false-negative results were obtained. Based on these values, the sensitivity, precision, and F1-score values were calculated to be 0.691, 0.478, and 0.565, respectively [38]. In another study using the U-Net framework, fully automatic segmentation of the computer-aided planning of orthognathic surgery orthognathic surgery planning was performed on CT images. In this research, the number of CT images totaled 454, which were separated into cohorts for training/validation (n = 300) and testing (n = 153). The Mean volumetric Dice Similarity Coefficient (vDSC) and surface Dice Similarity Coefficient at 1 mm (sDSC) were calculated for the test cohort, with values of 0.96 and 0.97 reported for the upper skull, 0.94 and 0.98 for the mandible, 0.95 and 0.99 for the upper teeth, 0.94 and 0.99 for the lower teeth, and 0.82 and 0.98 for the mandibular canal. Industry expert segmentation approval rates for the mandible, mandibular canal, upper skill, upper teeth and lower teeth were determined to be 93%, 89%, 82%, 69%, and 58%, respectively [39]. In another study that used a deep learning model based on a regression neural network to fully automate the process of segmenting airways using CBCT, 315 patient images were included. In this study, the analysis focused on the distinctions among the data measured using a manual process and data obtained via deep learning. Through the application of agreement analysis, the extraction of 61 samples was performed and then a comparison was made between the value obtained from the manual measurements and the value predicted by the deep learning network with respect to both coordinates and volumes. The intraclass correlation coefficient (ICC) that had the highest correlation was the total volume in the oropharynx (0.986), along with the hypopharynx (0.964), as well as the nasopharynx (0.912). The coordinate CV2(x) had the intraclass correlation coefficient (ICC) with the greatest correlation (0.963), whereas the lowest correlation was observed at CV4(y) (0.868) [40]. A comparison was made between the overall volume evaluated via deep learning and the measurements of the volume conducted utilizing regression analysis manually; the findings mirrored those of the current study in that the two measurements had slopes near to 1. In another study of orthognathic surgery patients using 160 whole skull CBCT scans (70 scans taken preoperatively and 90 taken postoperatively) using the 3D U-net algorithm of artificial intelligence, the mandible was segmented semi-automatically and fully automatically. On average, the time taken by the semi-automatic (SA) was 1218.4 s, while the time taken by the refined artificial intelligence (RAI) decreased significantly (*p* < 0.0001) to 456.5 s (2.7-fold decrease). According to the assessments of both inter- and intraoperator consistency, the performance of the RAI was superior to the SA for each of the metrics, suggesting that it was more consistent. Where the SA was taken as the ground truth, the intersection over union (IoU) score for the AI and RAI was 94.6% and 94.4%, respectively [41]. In our study, the automated parotid segmentation model was developed using U-Net architecture and deep learning techniques. The results of this study support that implementing such a system containing and not containing further user enhancements can maximize the efficiency, reduce human error, and provide more accurate predictions. In another retrospective study involving the segmentation of organs that utilized the U-Net AI algorithm, sample data were taken from individuals who had undergone a prostate MRI and ultrasound-MRI fusion transrectal biopsy in the period from September 2014 to December 2016. Two experts in abdominal radiology segmented axial T2-weighted images manually, which subsequently acted as the ground truth. Subsequent to the process of manual segmentation, the images were employed for training on a customized hybrid 3D-2D U-Net CNN architecture in a fivefold cross-validation paradigm for neural network training and validation. Statistical analysis was performed based on the Dice score, which measures the extent to which the segmentations performed manually and those derived automatically overlap, as well as the Pearson linear correlation coefficient of the prostate volume. A total of 299 MRI exams involving 298 patients were used to train the CNN (overall amount of MR images = 7774). The mean Dice score of the customized hybrid 3D-2D U-Net was 0.898 (range, 0.890–0.908), while the prostate volume had a Pearson correlation coefficient of 0.974 [42]. Similar to our study, this research showed that the 3D-2D U-Net CNN performed highly effectively in prostate segmentation and volumetric assessment application. Compared to the abovementioned studies, it can be thought that there are two main reasons why the values in our study were more positive. First, it can be considered that the parotid gland borders were easier to detect when compared to other structures. Secondly, it is possible that the repetitive checks during segmentation helped us to obtain more successful training data.

## 5. Conclusions

The findings of this study demonstrated that it is possible to use AI models based on deep learning to automatically segment the parotid gland on the axial CT images. Despite all these positive results, new studies with a significantly higher amount of training data and larger ROIs are required to distinguish the parotid gland from other anatomical structures.

## Figures and Tables

**Figure 1 diagnostics-13-00581-f001:**
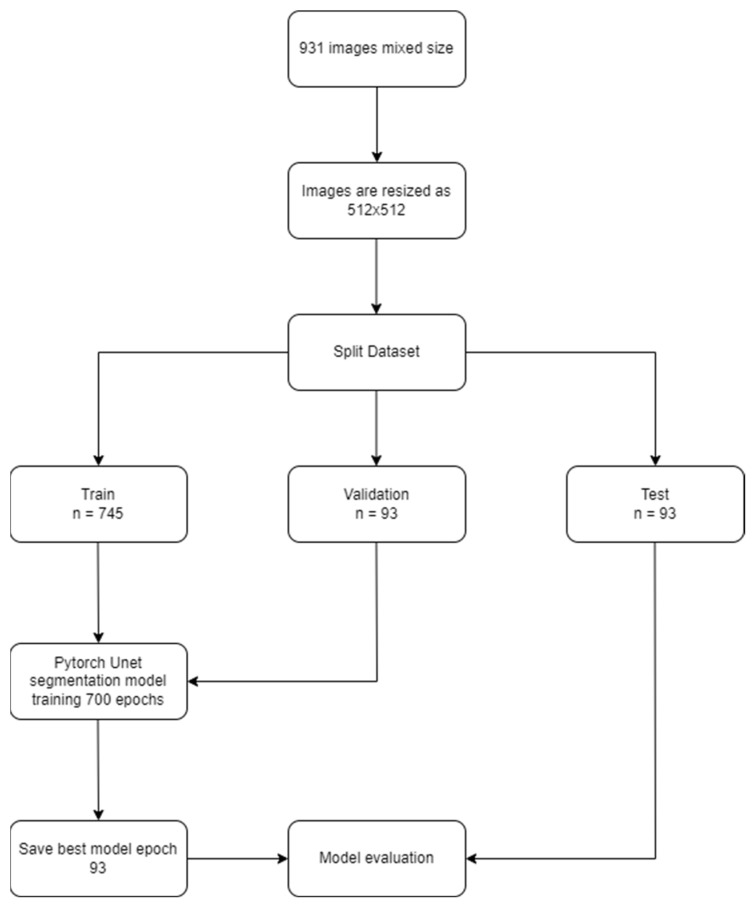
Model pipeline of parotid gland segmentation.

**Figure 2 diagnostics-13-00581-f002:**
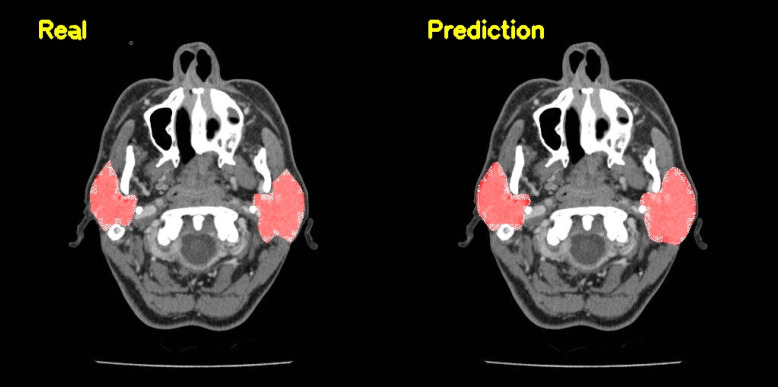
The automatic segmentation of the parotid gland utilizing the Artificial Intelligence model in axial CT slices.

**Figure 3 diagnostics-13-00581-f003:**
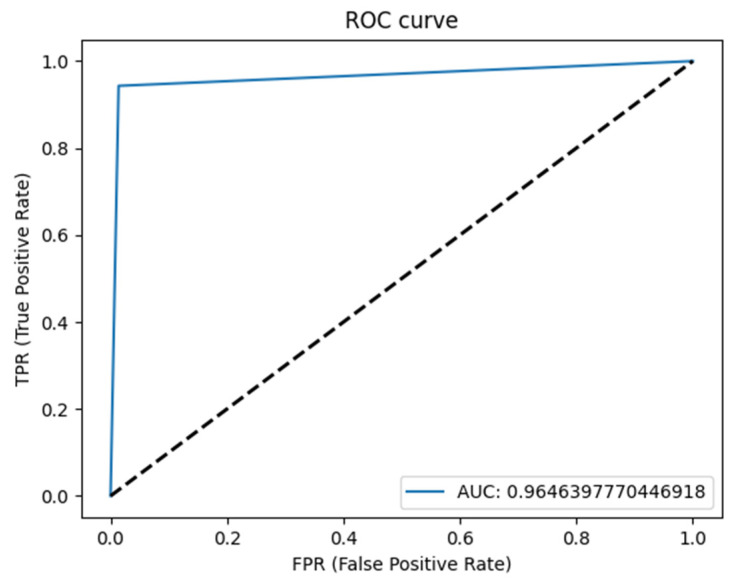
ROC curve.

**Figure 4 diagnostics-13-00581-f004:**
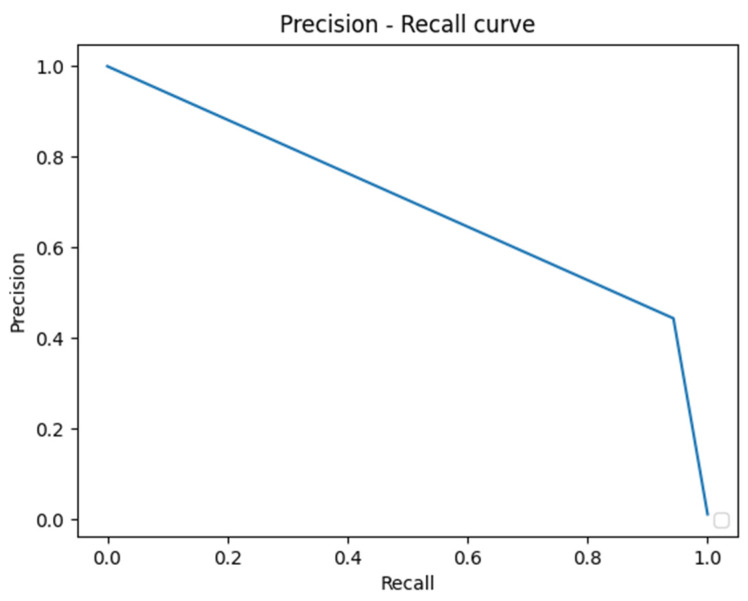
Precision–Recall curve.

**Table 1 diagnostics-13-00581-t001:** Results showing the predictive performance utilizing the AI model (CranioCatch, Eskisehir-Turkey) in terms of segmenting the parotid gland with the testing data.

Number	TP	FP	FN	Sensitivity	Precision	F1-Score
Sample	93	0	0	1.0	1.0	1.0
Label	184	0	0	1.0	1.0	1.0

## Data Availability

Not applicable.

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
