# Peer review of "Deep-Learning-Based Automatic Segmentation of Parotid Gland on Computed Tomography Images"

_diagnostics, 2023, doi:10.3390/diagnostics13040581_

Round 1
Reviewer 1 Report
- Literature review. The “deep learning methods” is not specific algorithm, and different algorithm varies in function and capability. However, there is no background description and discussion on which DL method should be used in this research.
- Method. I get no clue why they chose the U-Net.
- Experiment: How to determine the TP/FP/FN objects? According to pixel-by-pixel classification, or the overlapping parts between results and ground truth?
- Contribution. I recommend the authors highlight contributions and innovations to make readers easy to follow.
Other comments:
- check and correct keywords.
- line 67 - 71. Why repeat the statement in two sentences. Try to make the expression clean and clear.
- line 83. elaborate the term “many” with literature support.
- line 121 - 125. Apparently, the training/validation/test groups are not divided by 80/10/10.
Author Response
We would like to thank the reviewers and editors who contributed to the evaluation of our research. We think that your suggestions and comments improve our application. The revisions on the mentioned topics can be found below. In particular, detailed revisions were made in the introduction and method section. Also, you can find some other minor improvements colored in the new manuscript file.
// We shall thank the reviewer for his/her valuable comments and suggestions. The article is revised as requested when possible.
Literature review. The “deep learning methods” is not specific algorithm, and different algorithm varies in function and capability. However, there is no background description and discussion on which DL method should be used in this research.
// Literature review is added to the “Introduction”
Proceedings in digital imaging pave the way for implementing various artificial intelligence (AI) tools for segmenting, detecting and classifying the anatomical and pathological structures.[15, 16] Nowadays, the practice of radiology benefits significantly from AI applications. Implementing such tools can be highly beneficial in removing the burden of performing certain tasks repeatedly including segmenting organs or nerves or for extracting quantitative data that is more beneficial, thus enabling the clinician to increase their focus on attempting to solve complicated clinical issues.[17, 18] Still, there are still many problems regarding the efficient application of AI in radiologists’ normal practice. In the success of the developed AI model, the use of quality data in education and the correct labeling process are important.[19, 20]
Image segmentation is the subject of various fields such as transportation, architecture and medical imaging. Traditional segmentation methods, such as boundary extraction, threshold-based segmentation, and region-based segmentation can be adopted in manual segmentation of the medical images.[21, 22] However, manual segmentation requires expertise and is a time-consuming process. In the deep learning approach, features are extracted by algorithms by establishing multi-layered mathematical models. Thus, developers can benefit from the advantage of using big data in models training.[22, 23] Convolutional neural network (CNN) algorithms take attention with their success in image processing tasks. U-Net is an architecture developed for image segmentation. The basic structure consists of contracting and expansion paths, which are almost symmetrical, resulting in a u-like shape.[21-23] Deep learning can be utilized in U-Net algorithms, and its higher performance than its competitors in a limited dataset makes this architecture popular in segmentation tasks in the medical field where data is limited.[21]
Recently, the deep learning method has utilized extensively, especially in medical image processing where segmentation is needed.[16, 24] Segmentation from head and neck CT images was previously performed with the deep learning method.[25, 26] Yang et al. proposed a system based on atlas registration and support vector machine model for automated segmentation of the parotid gland using MR images in 2014. 15 patients with head and neck radiotherapy (42 MRI data) were included and the difference between the model and human tracings were reported 7.98% and 8.12% for the left and the right parotid, respectively.[27] In 2018, Močnik et al. developed an automatic multimodal method for segmentation of parotis glands from CT and MRI pair of patient data. Elastix and ANTs tools were employed to register MRI image to CT and the CNN model was implemented using Microsoft Cognitive Toolkit. The researchers compared results of the proposed multimodal model with the CT only modality, and reported a Dice overlapping coefficient of 78.8% for the first, and 76.5% for the latter approach.[28] Hänsch et al. developed a U-Net based system for segmenting parotid from CT images two-dimensional, three-dimensional and in a two-dimensional ensemble mode in 2019. 254 head and neck CT scans from two different clinical sites were selected, and in addition to the models’ performance for segmentation, number of the training samples were also investigated. The authors reported a mean Dice similarity of 0.83 for all three models, and increasing the training cases to more than 250 did not increased the Dice coefficient, significantly.[29]
Method. I get no clue why they chose the U-Net.
// Information about U-net algorithms is included in the ‘Introduction’.
U-Net is an architecture developed for image segmentation. The basic structure consists of contracting and expansion paths, which are almost symmetrical, resulting in a u-like shape.[21-23] Deep learning can be utilized in U-Net algorithms, and its higher performance than its competitors in a limited dataset makes this architecture popular in segmentation tasks in the medical field where data is limited.[21]
Experiment: How to determine the TP/FP/FN objects? According to pixel-by-pixel classification, or the overlapping parts between results and ground truth?
// Statistics section is detailed in ‘Methods’.
Statistics for the Model’s Performance
The model’s performance in automated segmentation of the parotid glands on axial CT images was evaluated with the F1-Score, the precision, the sensitivity, and the area under curve (AUC) values. The model’s result was considered successful if the prediction and the ground truth intersected over 50% in each individual image slice. True positive (TP), false positive (FP) and false negative (FP) were determined for calculating the performance metrics. The definitions and the formulas for model’s performance is calculated as described below:
True Positive (TP): At least 50%-pixels intersect between the automatic segmentation algorithm and the ground truth
False Positive (FP): At least 50%-pixels of the automatic segmentation algorithm do not intersect with the ground truth
False Negative (FN): At least 50%-pixels of the ground truth do not intersect with the results of the automatic segmentation algorithm
Contribution. I recommend the authors highlight contributions and innovations to make readers easy to follow.
// Aim of the study is revised in ‘Introduction’.
This study aims to develop a deep convolutional neural network (dCNN) algorithm based on U-Net architecture and evaluate the model’s performance in automatic segmentation of the parotid glands on axial-CT images.
Other comments:
- check and correct keywords.
// 2 more keywords are included:
U-Net, computed tomography
- line 67 - 71. Why repeat the statement in two sentences. Try to make the expression clean and clear.
// The manuscript is revised for a simpler explanation in the ‘Introduction’ and ‘Methods’ section.
- line 83. elaborate the term “many” with literature support.
// The sentence is detailed.
Still, there are still many problems, such as the need for large datasets and training, regulation issues, medicolegal responsibility, are suggested as the barriers for an efficient application of AI in radiologists’ normal practice.
- line 121 - 125. Apparently, the training/validation/test groups are not divided by 80/10/10.
// The number testing sample is revised in Methods and Table 1.
Data Split:
931 axial images were resized to 512x512 pixels. The data set was a separated into the training (80%), validation (10%), and test (10%) groups, randomly.
Training group: 745 (1445 labels)
Validation group: 93 (178 labels)
Test group: 93 (184 labels)
|
Number |
TP |
FP |
FN |
Sensitivity |
Precision |
F1 Score |
|
Sample |
93 |
0 |
0 |
1.0 |
1.0 |
1.0 |
|
Label |
184 |
0 |
0 |
1.0 |
1.0 |
1.0 |
_______________________________________________________________________

Reviewer 2 Report
The manuscript is well written. The introduction is very clearly and presents the objective of the study very well.
The materials, discussion and conclusion are well descrebed too. As cited the artificial intelligence is being increasingly to solve clinical probrems. The clinicians are enable to increase their focus on attempting to solve clinical issues
Author Response
We would like to thank the reviewers and editors who contributed to the evaluation of our research. We think that your suggestions and comments improve our application. The revisions on the mentioned topics can be found below. In particular, detailed revisions were made in the introduction and method section. Also, you can find some other minor improvements colored in the new manuscript file.
Reviewer 3 Report
The author carried out a relatively simple study. This paper explore the performance of automatic parotid gland segmentation model based on U-NET algorithm. The author used true positive, false positive, false negative, sensitivity, precision and F1 score as indicators. However, the study design is not rigorous enough, the subgroup analysis is lacking, and the presentation of results is not sufficient.
1)In the background, it is suggested that the author discuss the necessity of automatic segmentation in parotid gland, such as improving the consistency among observers and the repeatability of experimental results, saving labor and improving efficiency. Whether there is an automatic parotid gland segmentation model so far.
2)Author did not mention the inclusive and exclusive criteria of enrolled images.
3)How many images are included for each parotid gland ?
4)Are indicators such as true positive based on single slice? Or based on a single gland? Or based on a single patient? What's the definition of perfect match? 80% ?or 90%?
5)Has the author performed subgroup analysis, such as different sizes, different densities, different ages, different genders, different types of lesions,and different slice thicknesses and scanners?
6)Generally speaking, the parotid gland is composed of many pixels, and the possibility of different pixels being parotid gland or not is different (prediction score). Did the author analyze this?
7)Is there an external validation set? This will better verify the performance of the model
Author Response
The author carried out a relatively simple study. This paper explore the performance of automatic parotid gland segmentation model based on U-NET algorithm. The author used true positive, false positive, false negative, sensitivity, precision and F1 score as indicators. However, the study design is not rigorous enough, the subgroup analysis is lacking, and the presentation of results is not sufficient.
// We shall thank the reviewer for his/her valuable comments and suggestions. The article is revised as requested when possible.
1)In the background, it is suggested that the author discuss the necessity
of automatic segmentation in parotid gland, such as improving the consistency
among observers and the repeatability of experimental results, saving labor
and improving efficiency. Whether there is an automatic parotid gland
segmentation model so far.
// Previous studies related to parotid segmentation is included in the Introduction.
Yang et al. proposed a system based on atlas registration and support vector machine model for automated segmentation of the parotid gland using MR images in 2014. 15 patients with head and neck radiotherapy (42 MRI data) were included and the difference between the model and human tracings were reported 7.98% and 8.12% for the left and the right parotid, respectively.[27] In 2018, Močnik et al. developed an automatic multimodal method for segmentation of parotis glands from CT and MRI pair of patient data. Elastix and ANTs tools were employed to register MRI image to CT and the CNN model was implemented using Microsoft Cognitive Toolkit. The researchers compared results of the proposed multimodal model with the CT only modality, and reported a Dice overlapping coefficient of 78.8% for the first, and 76.5% for the latter approach.[28] Hänsch et al. developed a U-Net based system for segmenting parotid from CT images two-dimensional, three-dimensional and in a two-dimensional ensemble mode in 2019. 254 head and neck CT scans from two different clinical sites were selected, and in addition to the models’ performance for segmentation, number of the training samples were also investigated. The authors reported a mean Dice similarity of 0.83 for all three models, and increasing the training cases to more than 250 did not increased the Dice coefficient, significantly.[29]
[21] Du G, Cao X, Liang J, Chen X, Zhan Y. Medical image segmentation based on u-net: A review. JIST. 2020;64:1-12.
[22] Siddique N, Paheding S, Elkin CP, Devabhaktuni V. U-net and its variants for medical image segmentation: A review of theory and applications. Ieee Access. 2021;9:82031-57.
[23] Azad R, Aghdam EK, Rauland A, Jia Y, Avval AH, Bozorgpour A, et al. Medical image segmentation review: The success of u-net. arXiv preprint arXiv:221114830. 2022.
[24] Shen D, Wu G, Suk HI. Deep Learning in Medical Image Analysis. Annu Rev Biomed Eng 2017;19:221-48.
2)Author did not mention the inclusive and exclusive criteria of enrolled images.
// The inclusive and exclusive criteria is included in the Methods section.
Samples with clearly visible parotid glands bilaterally were included, while images with a gross anomaly and artifacts on the parotid gland were excluded.
3)How many images are included for each parotid gland ?
// On average, the answer is 931/30 = 31. Each image slice contains one or two parotid glands. However, this is an average number but there was no number restriction for determination of the borders of the parotid glands.
In this retrospective study, a total of 30 anonymized CT data were selected from the archive of the Radiology Department of the Faculty of Medical Sciences, University of Campinas (UNICAMP).
The resulting DICOM files were imported to Pydicom (https://pydicom.github.io/datasets) software, and in total, 931 axial-CT images with unilateral or bilateral appearance of parotid gland are exported in Joint Photographic Experts Group (JPEG) format.
4)Are indicators such as true positive based on single slice? Or based on a single gland? Or based on a single patient? What's the definition of perfect match? 80% ?or 90%?
// Statistics section in the Methods is revised. The model’s result was considered successful if the prediction and the ground truth intersected over 50% in each individual image slice (based on a single slice, with one or two parotids).
Statistics for the Model’s Performance
The model’s performance in automated segmentation of the parotid glands on axial CT images was evaluated with the F1-Score, the precision, the sensitivity, and the area under curve (AUC) values. The model’s result was considered successful if the prediction and the ground truth intersected over 50% in each individual image slice. True positive (TP), false positive (FP) and false negative (FP) were determined for calculating the performance metrics. The definitions and the formulas for model’s performance is calculated as described below:
True Positive (TP): At least 50%-pixels intersect between the automatic segmentation algorithm and the ground truth
False Positive (FP): At least 50%-pixels of the automatic segmentation algorithm do not intersect with the ground truth
False Negative (FN): At least 50%-pixels of the ground truth do not intersect with the results of the automatic segmentation algorithm
5)Has the author performed subgroup analysis, such as different sizes,
different densities, different ages, different genders, different types of
lesions,and different slice thicknesses and scanners?
// Subgroup analysis was not a part of our study design.
6)Generally speaking, the parotid gland is composed of many pixels, and the possibility of different pixels being parotid gland or not is different (prediction score). Did the author analyze this?
// Statistics section revised, as presented above.
7)Is there an external validation set? This will better verify the performance of the model
// External validation set was not included in the study design, yet 10% of the samples were split for validation.
_______________________________________________________________________
Academic editor’s comment:
Based on the 2 reviewer comments to “Reject”, I also feels to reject with
encourage resubmission because the manuscript is based on the simple U-Net
model and there is no comparison given with other techniques.
But I will let you have the final decision as the Editor.
// We shall thank the editor’s for his/her valuable comment. The article is revised and a literature review and more details are added. Unfortunately, comparing several techniques was not a part of our research, however, an algorithm (U-Net) popular in medical image segmentation was developed and evaluated. We hope that the paper is improved and can considered for publishing.
_______________________________________________________________________

Round 2
Reviewer 1 Report
My main concerns about this research are innovation and contribution. The authors claim that they aim to develop a novel U-net based DL method (line 118-120). Unfortunately, I was unable to find a logical argument to support either why U-Net outperformed other DL methods or the claim that CNN had "developed". According to the methodology section, the implemented Py-Torch library is being used without any development. Besides, to the best of my knowledge, there are several excellent studies on U-net based gland segmentation that share the similar hypothesis, but are not taken into consideration in this manuscript: - Zhao, Peng, et al. "SCAU-net: spatial-channel attention U-net for gland segmentation." Frontiers in Bioengineering and Biotechnology 8 (2020): 670. - Dabass, Manju, Sharda Vashisth, and Rekha Vig. "Attention-Guided deep atrous-residual U-Net architecture for automated gland segmentation in colon histopathology images." Informatics in Medicine Unlocked 27 (2021): 100784. - Du, Getao, et al. "Medical image segmentation based on u-net: A review." Journal of Imaging Science and Technology (2020). - Hänsch, Annika, et al. "Comparison of different deep learning approaches for parotid gland segmentation from CT images." Medical Imaging 2018: Computer-Aided Diagnosis. Vol. 10575. SPIE, 2018. Lastly, proofreading is helpful to improve this paper. Please correct/rewrite sentences like the one in lines 80-81.Reviewer 3 Report
I think the authors have done their best to revise the paper. I accept their changes. It is hoped that subgroup analysis can be performed to better interpret the performance of automatic parotid gland segmentation algorithm